# Cognitive Avoidance Is Associated with Decreased Brain Responsiveness to Threat Distractors under High Perceptual Load

**DOI:** 10.3390/brainsci13040618

**Published:** 2023-04-05

**Authors:** Vivien Günther, Mariia Strukova, Jonas Pecher, Carolin Webelhorst, Simone Engelmann, Anette Kersting, Karl-Titus Hoffmann, Boris Egloff, Hadas Okon-Singer, Donald Lobsien, Thomas Suslow

**Affiliations:** 1Department of Psychosomatic Medicine and Psychotherapy, University of Leipzig Medical Center, 04103 Leipzig, Germany; vivien.guenther@medizin.uni-leipzig.de (V.G.); carolin.webelhorst@medizin.uni-leipzig.de (C.W.); anette.kersting@medizin.uni-leipzig.de (A.K.); 2Department of Neuroradiology, University of Leipzig Medical Center, 04103 Leipzig, Germany; simone.mucha@medizin.uni-leipzig.de (S.E.); karl-titus.hoffmann@medizin.uni-leipzig.de (K.-T.H.); donald.lobsien@helios-gesundheit.de (D.L.); 3Department of Psychology, Johannes Gutenberg University of Mainz, 55122 Mainz, Germany; egloff@uni-mainz.de; 4Department of Psychology, University of Haifa, Haifa 3100000, Israel; hadasos@psy.haifa.ac.il

**Keywords:** avoidant coping, vigilant coping, repression, sensitization, fMRI, magnetic resonance imaging

## Abstract

Cognitive coping strategies to deal with anxiety-provoking events have an impact on mental and physical health. Dispositional vigilance is characterized by an increased analysis of the threatening environment, whereas cognitive avoidance comprises strategies to inhibit threat processing. To date, functional neuroimaging studies on the neural underpinnings of these coping styles are scarce and have revealed discrepant findings. In the present study, we examined automatic brain responsiveness as a function of coping styles using functional magnetic resonance imaging. We administered a perceptual load paradigm with contemptuous and fearful faces as distractor stimuli in a sample of *N* = 43 healthy participants. The Mainz Coping Inventory was used to assess cognitive avoidance and vigilance. An association of cognitive avoidance with reduced contempt and fear processing under high perceptual load was observed in a widespread network including the amygdala, thalamus, cingulate gyrus, insula, and frontal, parietal, temporal, and occipital areas. Our findings indicate that the dispositional tendency to divert one’s attention away from distressing stimuli is a valuable predictor of diminished automatic neural responses to threat in several cortical and subcortical areas. A reduced processing in brain regions involved in emotion perception and attention might indicate a potential threat resilience associated with cognitive avoidance.

## 1. Introduction

Anxiety is an emotional state that is characterized by high physiological arousal, vigilance, and the subjective experience of distress and apprehension, prompted by distant and unpredictable threats [1,2]. In general, anxiety is considered as an adaptive emotion that serves the maintenance of safety and well-being [3]. However, when feelings of anxiety become persistent and excessive, individuals may have a higher risk for cardiovascular diseases [4] or may even be diagnosed with an anxiety disorder when symptoms lead to functional impairments in daily life [5]. Coping strategies in response to stressful events have been implicated in the pathogenesis of mental and physical disorders (e.g., [6,7,8]). Therefore, coping skills are potential targets for prevention or intervention programs [9]. There are inter-individual differences in the dispositional use of coping strategies to deal with anxiety-inducing situations. Krohne [10] has suggested in his model of coping modes (MCM) for threatening situations two dimensions of dispositional coping behavior: vigilance and cognitive avoidance. Vigilance is characterized by an intense search for information related to the threatening event. In contrast, cognitive avoidance denotes the disposition to inhibit the processing of threatening information by directing attention away from threat cues that can be of external or internal nature. Factor analyses have confirmed the assumption that vigilant and cognitive avoidant strategies are moderately (negatively) related, but constitute independent dimensions of dispositional coping with anxiety [11,12]. In the MCM, Krohne [10] has claimed that anxiety-inducing incidents are characterized by the presence of aversive stimulation and ambiguity, which can trigger emotional arousal and experiences of uncertainty regarding the occurrence of threats. Vigilance is a coping strategy that is supposed to reduce uncertainty and the probability of bad surprises. Cognitive avoidance aims at the reduction of intense anxious arousal by shielding the individual from aversive stimulation. In the MCM, it is suggested that individuals with a low tolerance for experienced uncertainty are inclined to make use of a vigilant coping style. Thus, vigilant individuals monitor their environment intensely to be prepared for the occurrence of potentially threatening information. On the other hand, individuals that are highly susceptible for anxious arousal are supposed to avoid danger cues. Individuals scoring high in vigilance but low in cognitive avoidance have been designated as sensitizers. Individuals who consistently employ cognitive avoidance but not vigilance have been designated as repressors [11]. There is evidence for an overlap between the coping style classifications according to Krohne [10] and according to Weinberger et al. [13]; see Egloff and Hock [14] and Krohne et al. [11]. The simultaneous utilization of scales to assess social desirability and trait anxiety has been proposed by Weinberger et al. [13] to classify repression and sensitization based on median splits or cut-off scores. Within the framework of Weinberger et al.’s [13] theory of coping strategies, it is assumed that repressors report low scores in anxiety, as they avoid the awareness of their own anxiety, while they display high repressive defensiveness. In response to stress, repressors are characterized by discrepancies among low self-reported distress but high levels of autonomic arousal. Despite some differences in the theoretical foundations, Krohne’s [10] and Weinberger et al.’s [13] methods appear to assess related constructs, in particular with respect to repression and sensitization. Thus, findings based on both assessment procedures might be comparable [14], but see Kohlmann [15] for contradictory findings. The classificatory system of Weinberger et al. [13] is category-based, which has been criticized (e.g., [16]), whereas Krohne et al. [11] have provided with the Mainz Coping Inventory (MCI), a dimensional measure to assess cognitive avoidance and vigilance.

Neuroimaging research on vigilant and cognitive avoidant coping styles in the framework of Krohne [10] is scarce and revealed heterogeneous findings. Most studies have investigated brain activation as a function of coping styles during threat processing by comparing individuals categorized as repressors with individuals categorized as sensitizers. This categorical approach provided interesting insights regarding neural differences between repressors and sensitizers. During the passive viewing of visible or briefly displayed emotional faces, repressors have exhibited heightened responsivity to fearful faces in the anterior cingulate cortex (ACC) and in the temporo-occipital visual system, relative to sensitizers [17]. On the other hand, sensitizers have demonstrated exaggerated responsivity to angry faces in the frontal cortex and increased amygdalar reactivity in response to visible fearful faces. During an explicit emotion processing task, where very briefly presented facial emotions had to be labeled, repressors have shown stronger activity than sensitizers in frontal and temporo-parietal areas in response to angry and fearful faces [18]. Similar findings have been reported for the explicit threat-evaluation of visible emotional faces [19]. Here, repressors exhibited heightened activity in response to angry but not fearful faces in the temporo-parietal and frontal cortex, compared to sensitizers. Stronger activations in repressors may indicate an intensified processing of specific threatening stimuli, but may also represent increased attentional control or emotion regulation efforts to enable task performance (see also [17,18,19]). There is also evidence for stronger activations in repressors compared to sensitizers in the amygdala and insula during fear conditioning with electrical stimulation [20]. Higher activations in these emotion-related brain areas were accompanied by stronger responses in the ventral ACC and ventromedial PFC, brain regions that have been implicated in emotion evaluation, emotion-related learning, and emotion regulation [21,22]. In sum, studies using categorical approaches partly point to heightened reactivity in prefrontal, temporal, parietal, and occipital areas in repressors relative to sensitizers. Higher responsivity to threat appears to depend strongly on the emotional quality of the stimulus, as angry faces produce more homogeneous findings than fearful faces.

Vigilance appears to be positively related to trait anxiety, whereas cognitive avoidance shows a negative correlation with anxiety [11,15,23,24]. However, none of the reported imaging studies on coping styles have considered a potential influence of anxiety.

Given the background information that vigilant and cognitive avoidant strategies are not conceptualized as opposite poles of a continuum, but as independent dimensions of dispositional coping [11,12], it is surprising that only one study used a dimensional approach to predict brain responsiveness to emotional faces by both coping styles, each independently. Leehr et al. [25] have used in a large sample an automatic emotion processing task, where individuals had to match the identity of threatening or neutral faces. Thus, the emotional quality of the faces was not relevant for task performance. Here, no relation among cognitive avoidance and brain responsiveness to fearful and angry faces was observed. A cerebral hyper-responsiveness in high cognitive avoidance could not be confirmed. High vigilance was associated with diminished activity in the anterior cingulate gyrus in response to angry, but not fearful, faces. A recent study on brain structural correlates of vigilant and cognitive avoidant coping styles linked volumetric alterations in the thalamus to both strategies to manage anxiety-provoking situations [24]. The thalamus, along with the amygdala, has been implicated in an automatic alerting system when faced with potential threat-related cues [26], in which the thalamus sends coarse information of potential danger signals to the amygdala (e.g., [27,28]). Günther et al. [24] found high vigilance to be related to volumetric increases in the thalamus, whereas reduced thalamic gray matter volume was found in cognitive avoidance. The heterogeneity of previous findings might partly be attributed to different methodologies (e.g., extreme group comparisons or dimensional approaches) and variations in applied experimental tasks in functional magnetic resonance imaging (fMRI) studies (e.g., controlled or automatic emotion processing with varying stimulus durations). The inconsistent results point out the importance of further research.

Faces and their emotional content appear to be processed fast and efficiently by the human visual system [29,30]. In general, threat-related stimuli were shown to have an impact on improved behavioral control capabilities [31] and to activate a distributed brain network including the amygdala, insula, and inferior frontal and temporal gyrus; see [32] for an overview. There is evidence that emotional stimuli capture attention involuntarily [33,34] and that their valences are evaluated automatically [35]. However, the automatic processing of emotionally salient stimuli appears to depend on the availability of sufficient attentional resources (e.g., [36,37]). According to the Load Theory ([38,39], see [40] for an overview), the processing of task-irrelevant distractors is prevented when high task demands or a large amount of task-relevant information leave no spare perceptual capacity. In this case, attention to threat distractors is inhibited due to a lack of processing resources. Interestingly, brain responsiveness to emotional face distractors under different perceptual-load conditions has been shown to be modulated by non-clinical [37] and clinical anxiety [41].

To our knowledge, no previous study investigated the effects of attentional load on the neural processing of threat distractors as a function of dispositional coping styles. In the present study, functional magnetic resonance imaging (MRI) scans were selected from healthy individuals with varying degrees of vigilant and cognitive avoidant coping strategies while they performed a perceptual load task with contemptuous, fearful, and neutral faces. Fearful faces are considered as biologically salient stimuli that signal potential threats in the environment [42], whereas contempt has been characterized as a hostile emotion [43], which can imply disapproval and rejection [44]. Previous neuroimaging studies, which used a dimensional approach [24,25], pointed to associations between coping styles and brain functional and structural alterations in the ACC and the thalamus. Thus, the ACC and thalamus were chosen as our regions of interest (ROIs). We expected increased thalamic and reduced anterior cingulate responsiveness to threat faces in vigilance, and decreased thalamic reactivity in cognitive avoidance. Due to heterogeneous results from earlier studies with differing experimental designs, we used an exploratory whole-brain approach to examine the relationship between vigilant/avoidant coping and brain responsiveness under low and high perceptual load.

## 2. Materials and Methods

### 2.1. Participants

Forty-five healthy volunteers participated in this study. All participants had to be native German speakers with a normal or corrected-to-normal visual acuity. Participants were recruited via public notices in canteens and libraries and online advertisement in social networks. Exclusion criteria for study participation were a history of neurological or psychiatric diseases, MRI contraindications, head trauma involving loss of consciousness, and left-handedness. To exclude potential diagnoses of past or current Axis I disorders, the Structured Clinical Interview for DSM-IV Axis I disorders (SCID-I; [45]) was administered.

Two participants had to be excluded, as they were extreme scorers in overall hit rates (>4 *SD* below the mean), which was mainly due to omission errors indicating an inattentive task performance. The final sample included *N* = 43 subjects (22 women), with a mean age of 23.81 years (*SD* = 3.59) and a mean school education of 12.09 years (*SD* = 0.48). When all tasks were completed, participants received financial compensation.

### 2.2. Psychometric Measures

Cognitive avoidance and vigilance were assessed by the Mainz Coping Inventory (MCI; 11.23). The MCI is as a stimulus-response inventory that consists of 80 items. Eight anxiety-evoking scenarios are described, including ego threat (e.g., giving a speech) and physical threat (e.g., an encounter with dubious people at night), and for each scenario, five vigilant strategies, such as “anticipation of negative events” and “information search” (e.g., “I think about what questions might be asked after the speech” or “I watch the dubious people closely”) and five cognitive avoidant strategies, such as “re-interpretation” and “attentional diversion” (e.g., “I prefer to talk with friends about something other than the speech”) are administered. For each scenario, participants are instructed to state on a true-false scale which of the mentioned strategies they would prefer. Across all situations and items, sum scores are calculated separately for the vigilance and cognitive avoidance scales.

Levels of trait anxiety were measured with the trait version of the State-Trait Anxiety Inventory (STAI; [46]). Questionnaire characteristics of the final sample and correlations between the psychometric measures are presented in Table 1. Vigilance and cognitive avoidance showed a moderate negative relationship. Moreover, trait anxiety was positively correlated with vigilance and negatively correlated with cognitive avoidance.

### 2.3. fMRI Experiment: Perceptual Load Task

The automatic emotional face processing task, with a low and high perceptual load condition, was adapted from Bishop et al. [37]. Stimuli consisted of photographs of 24 actors (12 men) picturing either fear, contempt, or neutral facial expressions, chosen from the Radboud Faces database [47]. Six additional actors depicting neutral faces were used for an initial practice task. Subjects were told that they would perform a letter search task and were instructed to respond as accurately as possible. We were primarily interested in brain responsiveness to emotional expressions, not reaction times. Conscious awareness of an erroneous response appears to elicit activity in the ACC (e.g., [48]). To avoid interferences with error-related brain responsiveness, we aimed at a low number of overall error trials and did explicitly request accurate, but not necessarily fast, responses from the participants. Responses were given on a fiber optic response pad in each hand via button presses with the left and right index finger.

The duration of each trial was 3 s and it started with a fixation cross that was shown for 500 ms, followed by 1000 ms of face and letter presentation; see Figure 1. The emotional or neutral face stimulus was a task-irrelevant distractor that was superimposed by a string of six letters. The participants had to identify the target letters X and A. In the low perceptual load condition, the string was entirely composed of target letters (e.g., six As). In the high perceptual load condition, a single target letter was randomly mixed with non-target letters (T, H, U, M, W), increasing attentional search requirements. A gray screen where the assignment of the letter responses to the buttons was depicted followed for 1.5 s.

Depicted is the sequence of events within trials of the perceptual load task in the low load condition (top illustration) and high load condition (lower illustration). Distractor stimuli were contemptuous, fearful, or neutral faces. Targets were the letters X or A.

The task comprised 24 blocks (4 per condition) of 6 trials. Trials within a block were randomized with respect to the target letter, but were constant regarding the load condition (low vs. high) and facial emotional expression (fearful, contempt, neutral). We balanced the presentation frequencies of actors across the experiment. Each block lasted for 18 s and was followed by a 12 s blank screen. Two fixed counterbalanced sequences were chosen for the trials to avoid stimulus order effects. Presentation^®^ software was used for stimulus presentation and to record responses.

### 2.4. MRI Acquisition and Preprocessing

Structural and functional MR images were selected using a 3 T scanner (Magnetom Trio, Siemens, Erlangen, Germany) with a 20-channel coil. Structural images were obtained with a T1-weighted 3D MP-RAGE [49] with the following imaging parameters: TI 900 ms, TR 1900 ms, TE 2.65 ms, flip angle 9°, spatial resolution of 0.8 × 0.8 × 1 mm^3^, and two averages. Blood oxygen level dependent (BOLD) contrast sensitive images were collected using T2*-weighted echo-planar imaging (EPI) sequence (matrix 64^2^; resolution 3.5 × 3.5 × 3.5 mm^3^; TR 2.54 s; TE 30 ms; flip angle 90°; interleaved acquisition of 40 slices along the AC-PC plane; 237 images). To preprocess and analyze MRI data SPM8 (http://www.fil.ion.ucl.ac.uk/spm/, accessed on 14 November 2013) was used. The first four functional volumes were discarded to allow longitudinal magnetization to reach equilibrium. Preprocessing included motion-correction, slice time-correction, and co-registration. Anatomical images were segmented, including normalization to the Montreal Neurological Institute (MNI) template. The normalization parameters were then applied to the functional EPI series (resulting in a re-sampled voxel size of 3 × 3 × 3 mm^3^). A temporal high-pass filter (128 s) was applied. Functional data were smoothed (Gaussian kernel size = 6 mm).

### 2.5. Data Analyses

Mean accuracy was high with 97% (*SD* = 0.03%), and overall hit rates were not significantly correlated with the vigilance scale (MCI-VIG) or the cognitive avoidance scale (MCI-CAV), (*r*(41) = 0.25, *p* = 0.11 and *r*(41) = −0.07, *p* = 0.67, respectively). Coping styles were not related to effects of emotion on accuracy (hit rate neutral minus hit rate emotion) in the different perceptual load conditions (see Appendix A). A 2 (perceptual load condition: low vs. high) × 3 (emotion condition: fear, contempt, neutral) analysis of variance with repeated measures and hit rates as dependent variables revealed no significant main effects for load and emotion, or interactions (all *p*s > 0.21). Thus, hit rates were not influenced by the cognitive load condition or emotion condition, probably due to ceiling effects in task performance.

Functional MRI data were analyzed by modeling the onset and duration of 18 s for each block. Regressors were convolved with a hemodynamic response function for the six conditions (contempt-low load, contempt-high load, fear-low load, fear-high load, neutral-low load, neutral-high load). First-level *t*-contrasts were calculated for the low load condition by contrasting the contempt and fear condition with the neutral one, separately. The same was done for the high load condition, where contempt and fear were each contrasted with the neutral condition. These contrasts were chosen to determine whether activations were clearly triggered by the emotional content of facial expressions, under low and high load separately. First-level *t*-contrasts (and following second-level main effect analyses) were also calculated for the load factor, where the low load condition was contrasted to the high load condition of the same emotion (e.g., contempt under low load vs. contempt under high load). These contrasts allow conclusions about activations elicited by task demands under a certain emotional condition and are reported in Appendix A.

For the second-level analyses, one-sample *t*-tests were performed to determine main effects of contemptuous expressions (vs. neutral ones) under low load and under high load, and of fearful expressions (vs. neutral ones) under low load and under high load. Contrast images were entered into regression models with the respective individual MCI-VIG and MCI-CAV scores as regressors of interest. One regression model was calculated per coping style and per contrast. Due to the evidence that trait anxiety is associated with vigilant and cognitive avoidant coping strategies, STAI scores and sex were also entered in these models to regress out a possible influence.

Exploratory whole-brain analyses were conducted with a voxel-wise threshold at *p* < 0.001 (uncorrected) and an additional cluster-level threshold of *p* < 0.05, family-wise-error (FWE) corrected. ROI analyses were carried out for the bilateral thalamus and ACC. To create an anatomically defined mask, the WFU Pickatlas [50] was used (according to [51]). For ROI analyses, the statistical threshold was set to *p* = 0.05, FWE-corrected.

## 3. Results

### 3.1. Neural Main Effects

#### 3.1.1. Main Effects of Threatening Faces under Low Perceptual Load

Across all participants, contemptuous faces (compared to neutral faces) that were processed under low perceptual load conditions significantly evoked activation in the left superior and bilateral inferior parietal lobule (BA40), the right middle temporal and fusiform gyrus, the right inferior frontal gyrus (BA9), and the thalamus see Table 2. Fearful faces produced no significant brain activations relative to neutral faces under low perceptual load.

#### 3.1.2. Main Effects of Threatening Faces under High Perceptual Load

Contemptuous (vs. neutral) faces produced no significant brain activations under high load conditions. Fearful faces, compared to neutral faces, significantly evoked activations in the left inferior frontal gyrus, extending to the precentral gyrus, in the bilateral superior temporal gyrus, and the cerebellum; see Table 2.

### 3.2. Relationship among Brain Activation to Threat Faces and Coping Styles

#### 3.2.1. Vigilance

ACC and thalamus ROI-based analyses and whole-brain analyses showed no significant associations between a vigilant coping style and activations in response to contemptuous (vs. neutral) or fearful (vs. neutral) faces under low or high load conditions.

#### 3.2.2. Cognitive Avoidance

Exploratory whole-brain regression analyses and thalamus ROI-analyses with the cognitive avoidance scale did not reveal associations with brain reactivity to contemptuous (vs. neutral) or fearful (vs. neutral) faces under the low load condition.

In thalamus ROI-based regression analyses under high load conditions, cognitive avoidance was negatively correlated with responsiveness in the thalamus to contemptuous (vs. neutral) and fearful (vs. neutral) faces: peak voxel *xyz*: −6 −10 16, cluster size: 2, *T*-score = 3.79, *p_FWE_* = 0.03, and peak voxel *xyz*: −3, −13, 3, cluster size: 3, *T*-score = 4.25, *p_FWE_* = 0.01, each respectively. 

In exploratory whole-brain regression analyses for contemptuous faces in the high load condition (see Table 3 and Figure 2), cognitive avoidance showed a negative correlation with activity in large clusters including the left postcentral gyrus, extending to the bilateral precentral (BA6), middle and inferior frontal gyrus (BA8); the bilateral insula, and bilateral superior temporal gyrus; the bilateral lingual gyrus, parahippocampal gyrus, and calcarine sulcus (primary visual cortex); the precuneus, supramarginal and inferior parietal gyrus, and amygdala and thalamus. More clusters were revealed in the right anterior and middle cingulate gyrus (BA32), extending to the superior medial frontal cortex (BA8 and 9), in the bilateral supplementary motor area (BA6), extending to the medial frontal and middle cingulate gyrus, and also in the right postcentral gyrus, extending to the inferior parietal gyrus.

During the processing of fearful (vs. neutral) faces under high load, cognitive avoidance was significantly and negatively associated with brain activity in a large cluster including the right insula and subcentral gyrus, the superior temporal gyrus, and precentral gyrus, and bilateral posterior cingulate gyrus, thalamus, and bilateral lingual gyrus, cuneus, and precuneus. Additional clusters showing a negative correlation were also revealed in the postcentral gyrus, in the fusiform gyrus, and the bilateral anterior cingulate gyrus.

Following a reviewer’s suggestion, we calculated further regression analyses with the respective vigilance or cognitive avoidance scale as additional covariate to control for. The result pattern did not substantially change, but the overall number of activated voxels was slightly reduced. When additionally controlling for a potential influence of vigilant coping style, cognitive avoidance was no longer related to diminished activity in the anterior cingulate gyrus in response to fearful faces.

## 4. Discussion

With the present study, we aimed to examine the relationship between coping strategies in anxiety-provoking situations and automatic brain responsiveness to threatening facial expressions in case of low and high task demands. For the first time, a perceptual load task with contemptuous and fearful faces was administered in a non-clinical sample of young adults with varying degrees of vigilant and cognitive avoidant coping styles. The processing of threatening faces can be assumed to be automatic in our used paradigm, since the facial expressions were task-irrelevant distractor stimuli [52]. We aimed to shed light on coping styles as predictors of automatic brain reactivity to threat stimuli under low and high attentional load, after controlling for a potential influence of trait anxiety. In the low perceptual load condition, contemptuous faces, but not fearful faces, elicited activations in the parietal, temporal, and inferior frontal cortex. These brain regions have been previously shown to be activated in response to negative pictures during low perceptual load [53]. During low perceptual load, coping styles did not influence brain activity in response to contemptuous or fearful faces. Under high perceptual load, fearful faces, but not contemptuous faces, elicited increased activity in frontal and temporal areas, and in the cerebellum. Interestingly, temporal and frontal activation was also modulated by coping style. Individuals with a disposition to cognitively avoid anxiety-inducing information showed decreased brain reactivity to contempt and fear in several areas including the superior temporal gyrus, the anterior and posterior insula, the thalamus, the amygdala, the parieto-occipital lobe (supramarginal and inferior parietal gyrus, precuneus, cuneus), the anterior, middle and posterior cingulate gyrus, and pre- and postcentral gyrus. Additionally, reduced activations in cognitive avoiders have been observed during the processing of contemptuous faces in the superior and middle frontal and medial frontal gyrus. Despite a moderate negative correlation between vigilance and cognitive avoidance, only the avoidant coping style contributed to the prediction of automatic brain responsiveness to threat under high attentional load. Contrary to findings from Leehr et al. [25], our hypothesis of reduced anterior cingulate responsiveness in vigilance was not confirmed. Only individuals who tend to inhibit the processing of distressing external or internal stimuli and divert their attention away from them showed reduced activity in a widespread neural network implicated in emotion and attention. This reduced responsiveness in cognitive avoidance was only observed during high attentional load and was not influenced by trait anxiety.

As a subcortical brain structure, the thalamus is involved in the early visual encoding of salient stimuli and plays an important role in the selection of information from the environment that can be relevant for further processing [54,55]. The thalamus has connections with the prefrontal and temporo-parietal cortices and with the amygdala [56,57]. The amygdala plays a key role in the fast detection of potential threats, and is involved in the recruitment of attentional resources and guidance of attention in the presence of salient stimuli [54,58,59]. A subcortical route has been suggested through which the amygdala receives direct sensory input from the thalamus [27,54,57]. In contrast, Pessoa and Adolphs [60] have proposed that the thalamus receives multimodal information from cortical regions and conveys biologically significant information to cortical regions, which activate the amygdala.

A previous study demonstrated a reduced thalamic gray matter density in cognitive avoidance [24]. Reduced thalamic and amygdalar responsiveness in individuals scoring high in cognitive avoidance may indicate a lower sensitivity to distracting threat stimuli and a lower capacity for the maintenance of attention to these stimuli. Of note, in the present study, findings in the thalamus were restricted to a relatively small cluster and should be considered as preliminary.

High cognitive avoidance was associated with reduced reactivity in visual processing areas (i.e., cuneus, lingual gyrus, see [61]) that have also been implicated in emotional face processing [62,63]. Diminished reactivity to threat distractors in cognitive avoiders was also observed in the inferior parietal cortex and supramarginal gyrus. These parietal brain areas appear to be involved in emotion regulation, such as reappraisal and suppression [22,64,65], probably by directing attention to relevant stimuli (e.g., [64]). Indeed, the ventral regions of the parietal cortex along with the precuneus (e.g., [66]) have been suggested to play a role in visuo–spatial attention [67], particularly in involuntary and bottom-up driven shifts of attention [68]. This proposed function is in line with increased activity in these brain areas during the non-conscious or automatic perception of fearful faces [69]. Additionally, the supramarginal gyrus appears to be recruited during the judgment of facial emotions [70]. Further, the superior temporal gyrus is supposed to be involved in the visual analyses and encoding of emotional facial expressions (e.g., [30,62,71]), and activity within this area has been linked to fear and anxiety [32].

It is conceivable that the visual processing and encoding of threat-related distractors and their capability to capture attention under high load is diminished in cognitive avoidance. Blunted responsiveness in the pre- and postcentral gyrus (i.e., the premotor and primary somatosensory cortex) may also indicate a reduced perception and recognition of threat faces in avoiders, since both brain regions have been implicated in the affective evaluation of facial expressions [30,71]. We have also observed in cognitive avoidance a reduced reactivity in the posterior cingulate gyrus. The posterior cingulate gyrus has been discussed as part of the default mode network [72] and appears to be involved in the evaluation of the affective valence of stimuli [73], in self-related processing, and in social evaluation (e.g., see [74] for an overview). Particularly, deactivations in the posterior cingulate gyrus have been described as a neural correlate of task-related attention [72]. Thus, decreased responsivity may indicate a more external focus on task demands and more efficient cognitive functioning in individuals with a disposition to cognitively avoid anxiety- inducing stimuli. However, no evidence emerged in our study that coping styles were associated with improved or impoverished task performance in case of low or high perceptual load. Thus, a more efficient processing style and a reduced encoding of threat distractors on the neural level in avoidant coping style did not exert an influence on the behavioral performance.

Individuals who scored high in cognitive avoidance also responded less to contemptuous and fearful faces in the posterior insula. The posterior insula has been ascribed to various functions, such as sensory-motor processing and the collection of interoceptive, emotional, and environmental data [75]. Menon and Uddin [76] have proposed a basic function of the posterior insula in the regulation of physiological reactivity, but it has also been suggested that the posterior insula engages in primary interoceptive processes [77,78] and in the representation of aversive bodily and emotional stages [79]. Activity within the posterior insula has also been associated with the judgment of emotions in faces [80] and attention to the emotional relevance of stimuli [81]. In addition, contemptuous faces elicited less activation in cognitive avoiders in the anterior insula, which has been implicated in attention, salience evaluation, emotional awareness, and in the integration of interoceptive information [75,77]. Reduced insular responsiveness in high cognitive avoidance may represent a less internally orientated processing of emotional or bodily states, which were induced by threat faces, or reduced attention to their relevance. This interpretation falls in line with reduced activity in the posterior cingulate and supramarginal gyrus, which have also been implicated in attention to one’s own emotional states [82].

Another brain region where individuals with a cognitive avoidant coping style manifested less activity in response to contemptuous faces included the dorsal anterior and middle cingulate gyrus, and the dorsomedial prefrontal cortex (PFC). The dorsal ACC and dorsomedial PFC have been suggested to support attention processes and performance monitoring and to play a role in voluntary emotion regulation [83] and conflict monitoring [84]. There is also convincing evidence that the dorsal ACC and dorsomedial PFC are engaged in fear appraisal and the generation of fear responses (e.g., [84,85]), in the evaluation of negative valences [86], and the anticipation of unpredictable threats [87]. Earlier findings point to a role of the dorsal ACC in subliminal face processing [88] and in cardiovascular responsivity to emotional face stimuli [89] and stressors [90]. Higher activity in the dorsal ACC and dorsomedial PFC has been linked to the processing of disapproving facial expressions in individuals with high rejection sensitivity [91] and to the experience of distress during social rejection [92]. Thus, individuals with a disposition to avoid threat-related information might demonstrate a lower sensitivity and autonomic responsiveness to facial expressions of contempt, which imply disapproval and rejection [44]. It is also conceivable that reduced activity in these brain areas indicates less effort to regulate emotion perception or to monitor emotional conflicts. Individuals with an avoidant regulation style might show an impaired ability to disengage attention and regulate the processing of task-irrelevant threat distractors under high load. However, in this case, one would expect heightened activations in emotion-related brain areas to accompany decreased emotion regulation efforts. An alternative interpretation of reduced ACC and PFC responsiveness to contempt might be a lower demand for emotional conflict monitoring or emotion regulation due to the diminished processing of threat distractors in subcortical and primary visual areas in cognitive avoidance. 

Taken together, according to our results, individuals who are inclined to withdraw their attention from anxiety-inducing information and to inhibit their further processing manifest decreased activity in a brain network implicated in the processing of emotionally relevant information. Our findings suggest reduced visual processing and encoding of distracting and threatening social stimuli, and less attention allocation and weaker emotional responsiveness to them. However, this reduced sensory analysis of threat signals and diminished emotional responsiveness in cognitive avoidance was only observed during high attentional load. Leehr et al. [25] have found no relationship among cognitive avoidant coping style and automatic brain responsiveness to emotional faces in a task with relatively low cognitive demands. This is in line with our null results in the low perceptual load condition. These data suggest that coping styles only modulate the processing of threatening information when perceptual resources are limited. One may speculate that coping-related inter-individual differences in threat encoding occur at a very early pre-attentive level, with a reduced distractor encoding in low (relative to high) cognitive avoidance at this processing stage. However, under the condition of spare attentional capacity (i.e., low load), salient distractors may be encoded at later processing stages in all individuals, irrespective of their dispositional use of coping strategies. Our results appear contradictory to findings by Paul et al. [18] and Rauch et al. [19], which pointed to enhanced processing of threat faces in prefrontal and temporo-parietal areas in repressors, relative to sensitizers. Several methodological differences might have contributed to the discrepancies. Importantly, earlier tasks required the participants to attend to and evaluate emotional expressions, whereas in our task emotional expressions served as task-irrelevant distractor stimuli. Dimensional approaches to investigate neural correlates of coping styles might also produce different results than category-based approaches where high scorers in cognitive avoidance and vigilance are compared to each other. In general, findings on functional substrates of strategies to cope with anxiety-provoking situations remain inconsistent and highlight the importance of future research. Further investigations appear worthwhile, since coping styles can have important implications for physical and mental health.

Schwerdtfeger and Rathner [93] suggested that repression constitutes a rather maladaptive strategy to cope with aversive events. According to the authors, a strategy to deny problems that require a solution or to divert one’s attention away from them may circumvent adaptive problem-focused approaches. Grant et al. [94] have provided evidence for a relationship among cognitive avoidance in response to stressful life events and anxiety. However, Coifman et al. [95] and Bonanno et al. [96] have reported protective effects of repressive coping on mental health, when individuals are confronted with adverse life events. It has been stated that cognitive avoidance serves the purpose of decreasing anxious arousal and should be preferred by people who have a low tolerance for intense emotional states [97]. Indeed, some studies have suggested a higher sensitivity and autonomic responsivity in the face of threats in repression (e.g., [20,98,99]), but other studies could not confirm this assumption [100,101]. It appears that repression and cognitive avoidance in response to stressful events are broadly defined constructs in the literature, comprising strategies such as denial, resignation, avoiding thoughts about problems, and avoiding the awareness of one’s own anxiety, but also attentional diversion, re-interpretation, and accentuating positive aspects of the situations [13,97,102]. These constructs vary significantly in their operationalization. In several studies, different coping questionnaires [20,94,98] were used, or repression has been operationalized as a discrepancy score between self-reported perception of distress and actual physiological responses (e.g., [95,99]), or as a combination of low self-reported trait anxiety and high defensiveness [100,101]. Within Krohne’s [10] MCM, cognitive avoidance and vigilance strategies are elaborated in the context of anxiety-inducing events with varying degrees of controllability. Cognitive functions that are subsumed under cognitive avoidance are, for example, attentional diversion, re-interpretation, denial, minimization, and self-enhancement (see [97]). Importantly, cognitive avoidance according to Krohne [10] is conceptually distinct from (behavioral) avoidance of anxiety-inducing situations, which is a common symptom in clinical anxiety that perpetuates the disorder [5]. Vigilance comprises strategies such as information search, anticipation of negative events, planning, and situation control [97]. It is plausible that the success of each strategy and its long-term consequences on physical and mental health depend on the degree to which threat situations can be controlled. Cognitive avoidance has been suggested as an appropriate strategy when dealing with uncontrollable threats [97]. In contrast, individuals who employ a vigilant coping style are inclined to have an intolerance of uncertainty, which has been suggested as a transdiagnostic causal mechanism of anxiety-related conditions [103].

Whereas earlier findings pointed to enhanced processing of emotional faces in a frontal and temporo-parietal brain network in healthy individuals with a disposition to employ a cognitive avoidant coping style relative to sensitization (e.g., [18]), our results indicate a diminished capacity to perceive and attend to threat distractors in cognitive avoidance. This relationship was not influenced by trait anxiety. Thus, dispositional strategies to cope with anxiety-evoking situations and anxious arousal in a cognitive avoidant manner can explain incremental proportions of variance in automatic brain activity elicited by threat faces.

Our findings of reduced activation in cognitive avoidance might have relevance within the framework of non-clinical and clinical studies on anxiety. Increased activation in the dorsolateral PFC, posterior cingulate, parietal, and temporo-occipital areas have been implicated in an anxious temperament and in social anxiety disorder [104,105,106,107]. Furthermore, in contrast to healthy controls, patients with panic disorder, specific phobias, and social anxiety disorder appear to exhibit elevated activity within the anterior and midcingulate gyrus in response to feared stimuli [108,109,110]. This hyperactivity was shown to be resolved after psychotherapy [110]. Our finding of reduced automatic brain responsiveness to negative distractor stimuli might indicate a more favorable processing style in cognitive avoidance and might be a resilience factor. However, it remains to be clarified in future studies whether a cognitive avoidant strategy that relies on attention allocation away from anxiety-inducing situations and from bodily responses to them, as well as on denial, minimization, reappraisal, and the accentuation of positive aspects of a threat situation can be clearly assigned as an adaptive or maladaptive way of coping. It remains speculative whether cognitive avoidance increases the risk for physical and mental health issues or can be considered as a protective factor. Our results indicate a neural response pattern to emotional stimuli that appears incongruous to brain activations that have been observed in pathological anxiety.

Our study has several limitations. We have investigated young, healthy, and well- educated participants. Our results cannot be generalized to other populations. In our experiment, 24 trials were presented per condition. An increased trial number may enhance the statistical power. However, one has to consider potential neural habituation effects with an increased task length and repetition of stimuli [111]. With a sample size of *N* = 43 we were only able to detect moderate and large effects. To test their validity, our findings need to be replicated in larger samples and merged in meta-analyses. We operationalized vigilant and cognitive avoidant coping by means of the Mainz Coping Inventory [11,23]. Future studies on physiological or neural responsiveness and health outcomes that are moderated by coping styles might benefit from the simultaneous application of different measures and operationalization of cognitive avoidance. The cross-sectional design of our study does not allow speculations about the direction of the effects. Longitudinal studies may investigate whether altered neural reactivity is a mechanism in the manifestation of inter-individual differences in coping styles or might be the consequences of down-regulated perception and attention in individuals who prefer cognitive avoidant anxiety regulation styles in their daily lives.

## 5. Conclusions

In sum, we have demonstrated in individuals with a disposition to cognitively avoid anxiety-inducing information reduced reactivity to distracting threat faces in a widespread subcortical and cortical brain network that is involved in attention and emotion perception. 

It might be concluded from our findings that individuals who employ a cognitive avoidant regulation style have a less sensitive threat detection system when attentional resources are limited. The diminished threat processing may indicate a potential resilience against transient danger signals in the environment and may be a neural basis of inter-individual differences in the vulnerability to mental disorders.

## Figures and Tables

**Figure 1 brainsci-13-00618-f001:**
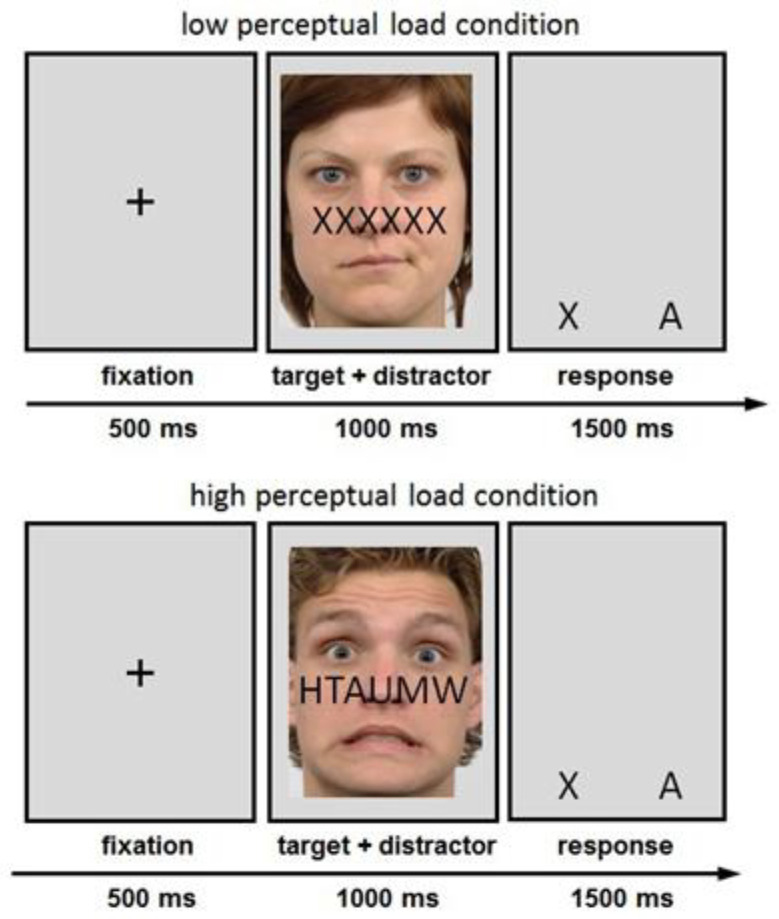
FMRI paradigm.

**Figure 2 brainsci-13-00618-f002:**
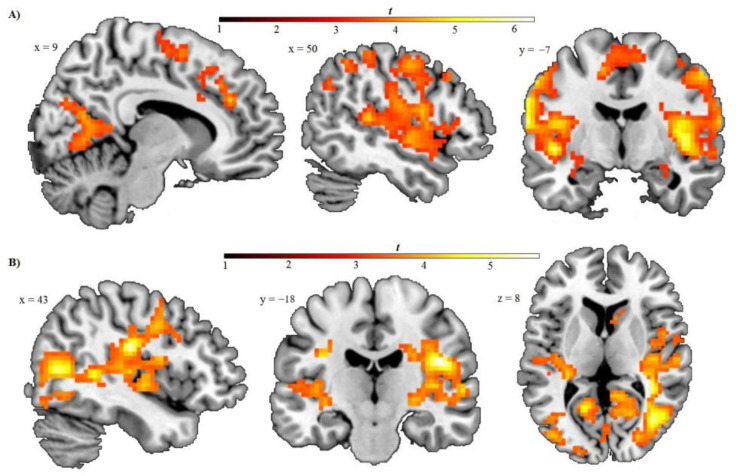
Results from whole-brain regression analyses with cognitive avoidance predicting decreased brain responsiveness to contemptuous and fearful faces under high load. (**A**) Sagittal and coronal images in neurological orientation showing the negative relationship between cognitive avoidance and responsivity to contemptuous (vs. neutral) faces under high perceptual load. Depicted are activation clusters in the right superior medial frontal and cingulate gyrus (BA32), supplemental motor area (BA6), precuneus, cuneus, and lingual gyrus (top left), in the right precentral (BA4), inferior frontal gyrus, and parietal and superior temporal lobes (top middle), and in the insula, postcentral gyrus, superior temporal gyrus, left amygdala, and supplementary motor area (top right). (**B**) Sagittal, coronal, and axial images showing the negative correlation between cognitive avoidance and activation in response to fearful (vs. neutral) faces under high perceptual load. Depicted are activation clusters in the precentral, supramarginal, and superior temporal gyrus (lower left), in the insula, subcentral gyrus, and superior temporal gyrus (lower middle), and in the posterior insula, superior temporal gyrus, calcarine, and middle occipital gyrus (lower right). The voxel-wise threshold was set to *p* = 0.001 (uncorrected) with a cluster-level threshold of *p* < 0.05, FWE-corrected. Color bar: *t*-values.

**Table 1 brainsci-13-00618-t001:** Descriptive statistics and correlations (*df* = 41) between psychometric measures.

		Coping	Trait Anxiety
	*M* (*SD*)	MCI-CAV	STAI
MCI-VIG	22.00 (6.80)	−0.41 **	0.35 *
MCI-CAV	23.75 (5.11)	-	−0.36 *
STAI-trait	35.05 (6.82)	-	-

* *p* < 0.05; ** <0.01 (two-tailed) MCI = Mainz Coping Inventory; VIG = Vigilance scale, CAV = Cognitive avoidance scale; STAI = State-Trait Anxiety Inventory.

**Table 2 brainsci-13-00618-t002:** Threat main effects: Brain regions showing increased activation in response to threatening (vs. neutral) faces under low and high perceptual load.

	Hemisphere	Peak *T*-Value	Peak-Level	Cluster	Cluster-Level	Peak MNI		
			*p* _uncorrected_	Size (Voxels)	*p_FWE_*	*x*	*y*	*z*
Low perceptual load	-	-	-	-	-	-	-	-
Contempt > neutral	-	-	-	-	-	-	-	-
Superior extending to inferior parietal lobule	R	5.61	<0.001	183	0.004	42	−55	55
Middle temporal extending to fusiform gyrus	R	5.23	<0.001	536	<0.001	57	−46	−5
Inferior frontal gyrus	R	5.33	<0.001	278	0.003	45	26	22
Inferior parietal lobule (BA40)	L	5.10	<0.001	183	0.004	−42	−52	55
Inferior temporal gyrus	L	4.57	<0.001	171	0.005	−42	−49	−17
Thalamus (ROI)	L	3.69	<0.001	1	0.04	9	−28	1
Fear > neutral	-	-	-	-	-	-	-	-
-	-	-	-	-	-	-	-	-
High perceptual load	-	-	-	-	-	-	-	-
Contempt > neutral	-		-	-	-	-	-	-
-	-	-	-	-	-	-	-	-
Fear > neutral	-		-	-	-	-	-	-
Inferior frontal extending to precentral gyrus	L	4.73	<0.001	230	0.003	−51	17	19
Superior temporal gyrus	R	4.89	<0.001	267	0.001	−57	−43	13
Superior temporal gyrus	L	5.33	<0.001	217	0.004	−57	−46	13
Cerebellum	R	4.62	<0.001	419	<0.001	42	−67	−29

Neuroanatomical labels, hemisphere, peak voxel *t*- and *p*-values, cluster extent, cluster-level *p*-values, and coordinates in MNI space are presented. Activation clusters are yielded by one-sample *t*-tests and are significant at *p* = 0.001 (uncorrected) and a cluster-level threshold of *p_FWE_* < 0.05.

**Table 3 brainsci-13-00618-t003:** Brain regions showing a negative correlation with cognitive avoidant coping style in response to threatening faces under high perceptual load.

	Hemisphere	Peak *T*-Value	Peak-Level	Cluster	Cluster-Level	Peak MNI		
			*p* _uncorrected_	Size (Voxels)	*p_FWE_*	*x*	*y*	*z*
Contempt (vs. neutral)	-	-	-	-	-	-	-	-
Left postcentral gyrus, extending to the following regions:	L/R	6.29	<0.001	5770	<0.001	−63	−10	16
Bilateral precentral (BA6), superior, and middle frontal gyrus (BA8)	L/R	5.18	<0.001	-	-	−51	−1	16
Bilateral posterior and anterior insula	L/R	6.14	<0.001	-	-	42	−4	4
Bilateral lingual and parahippocampal gyrus, calcarine sulcus and precuneus	L/R	5.16	<0.001	-	-	−18	−55	1
Bilateral superior temporal gyrus	L/R	5.38	<0.001	-	-	−48	−37	19
Bilateral supramarginal and inferior parietal gyrus	L/R	4.67	<0.001	-	-	−63	−19	37
Amygdala	L/R	3.46	<0.001	-	-	−27	−7	−11
Thalamus	L	3.79	<0.001	-	-	−6	−10	16
Dorsal anterior and middle cingulate gyrus (BA32), extending to medial frontal cortex (BA6,8,9)	R	4.59	<0.001	214	0.002	9	35	28
Bilateral supplementary motor area (BA6), extending to the left medial frontal and middle cingulate gyrus (BA31)	L/R	4.51	<0.001	221	0.001	−12	−7	52
Right postcentral gyrus extending to inferior parietal gyrus	R	4.13	<0.001	158	0.008	33	−40	58
Fear (vs. neutral)								
Subcentral gyrus, extending to the following regions:	L/R	5.67	<0.001	3634	<0.001	45	−19	22
Bilateral superior temporal gyrus	L/R	4.69	<0.001	-	-	45	−10	−2
Posterior insula	R	4.81	<0.001	-	-	45	−10	1
Precentral gyrus	R	4.68	<0.001	-	-			
Bilateral posterior cingulate	L/R	3.77	<0.001	-	-	12	−40	28
Bilateral lingual gyrus, cuneus and precuneus	L/R	5.05	<0.001	-	-	15	−43	−5
Amygdala	R	3.49	<0.001	-	-	27	−4	−11
Thalamus	L/R	4.25	<0.001	-	-	−3	−13	−2
Bilateral anterior cingulate gyrus (BA25, 32)	L/R	4.75	<0.001	353	0.001	−3	17	−5
Fusiform gyrus	L	4.22	<0.001	307	<0.001	−36	−46	−23
Postcentral gyrus	R	4.49	<0.001	161	0.008	12	−43	73
Postcentral extending to superior temporal gyrus	L	4.41	<0.001	110	0.03	−36	−19	28

Neuroanatomical labels, hemisphere, peak voxel *t*- and *p*-values, cluster extent, cluster-level *p*-values, and coordinates in MNI space are presented. Activation clusters are yielded by regression analyses and are significant at *p* = 0.001 (uncorrected) and a cluster-level threshold of *p_FWE_* < 0.05.

## Data Availability

The datasets used and/or analyzed during the current study are available from the corresponding author on reasonable request.

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
