# Peer review of "Cognitive Avoidance Is Associated with Decreased Brain Responsiveness to Threat Distractors under High Perceptual Load"

_brainsci, 2023, doi:10.3390/brainsci13040618_

Round 1
Reviewer 1 Report
The manuscript is well written with detailed background and clear methods and results. I have 3 inquiries/suggestions:
1) For the first fMRI results contrasting threatening emotions in separate perceptual loads, thalamic activation was reported and briefly discussed, but the relevant summary table reported only 1 voxel in the 'cluster'. If it's not a typo, I suggest not to discuss this, although thalamic involvement would have been nice. Or at least have a note of caution.
2) Can the authors elaborate a bit more on why the avoidance-related effects only showed up in high perceptual load but not low load in the discussion? High perceptual load intuitively would fully tax an individual's perceptual resources, I would expect individual differences in cognitive styles on perceiving/encoding threatening emotions to be less likely?
3) Did the authors also consider including both dimensions of avoidance and vigilance in one regression model and see what neural correlates are related to one dimension 'controlling' for the other? Or why not?
Author Response
General remarks: The manuscript is well written with detailed background and clear methods and results. I have 3 inquiries/suggestions:
Comment 1: For the first fMRI results contrasting threatening emotions in separate perceptual loads, thalamic activation was reported and briefly discussed, but the relevant summary table reported only 1 voxel in the 'cluster'. If it's not a typo, I suggest not to discuss this, although thalamic involvement would have been nice. Or at least have a note of caution.
Our response: The reviewer is right, that a cluster extent of 1 voxel for the FWE-corrected ROI analyses in the thalamus can be an irrelevant finding. We did not define a cluster size threshold for ROI analyses. Therefore, we did not discard this brain region from the table. However, to avoid an overestimation of this finding, we removed the thalamus from our results summary of main effect in the discussion [p. 11] and highlight in our revised text [p. 11, 3rd paragraph] that findings in the thalamus may be considered with caution due to the small cluster size.
Comment 2: Can the authors elaborate a bit more on why the avoidance-related effects only showed up in high perceptual load but not low load in the discussion? High perceptual load intuitively would fully tax an individual's perceptual resources, I would expect individual differences in cognitive styles on perceiving/encoding threatening emotions to be less likely?
Our response: The reviewer raised an interesting question. We provide a speculation on our result pattern in our revised discussion [p. 13, 2nd paragraph].
Comment 3: Did the authors also consider including both dimensions of avoidance and vigilance in one regression model and see what neural correlates are related to one dimension 'controlling' for the other? Or why not?
Our response: We agree with the reviewer that the inclusion of vigilance or cognitive avoidance as a control variable can be an interesting addition to the regression results. We conducted the suggested analyses, but our results did not change substantially. The overall cluster size was slightly reduced, several peak voxels minimally moved their position, and activation clusters in the anterior cingulate gyrus disappeared for the fearful face contrast. We provide the respective information in our revised results section [p. 10]. Due to minor changes in statistical values and brain areas, and due to a strict deadline for our revision, we would refrain from providing detailed tables with t/p-values, cluster sizes and MNI coordinates of these additional regression analyses results in our supplement.
Reviewer 2 Report
The present research article by Günther, entitled ‘Cognitive avoidance is associated with decreased brain responsiveness to threat distractors under high perceptual load’ is a well-written and useful summary on the current status of knowledge on neural underpinnings of copying strategies to inhibit threat processing.
The main strength of this manuscript is that it addresses an interesting and timely question, describing how the tendency to divert one’s attention away from distressing stimuli may be a valuable predictor of diminished automatic neural responses to threat in several cortical and subcortical areas.
In general, I think the idea of this article is really interesting and the authors’ fascinating observations on this timely topic may be of interest to the readers of Brain Sciences. However, some comments, as well as some crucial evidence that should be included to support the author’s argumentation, needed to be addressed to improve the quality of the manuscript, its adequacy, and its readability prior to the publication in the present form, in particular reshaping parts of the Introduction and Methods sections by adding more evidence and theoretical constructs.
Please consider the following comments:
• Abstract: According to the Journal’s guidelines, the abstract should be a total of about 200 words maximum. Please correct the actual one.
• A graphical abstract that will visually summarize the main findings of the manuscript is highly recommended.
• In general, I recommend authors to use more references to back their claims, especially in the Introduction of this article, which I believe is lacking. Thus, I recommend the authors to attempt to expand the topic of their article, as the bibliography is too concise. Nevertheless, I believe that less than 50 articles are too low for a research article. Therefore, I suggest the authors to focus their efforts on researching relevant literature: in my opinion, adding more citations will help to provide better and more accurate background to this study.
• Introduction: The ‘Introduction’ section is well-written and nicely presented, with a good balance of descriptive text and information about neural processing of threating emotions. Nevertheless, I believe that more information about neural bases of emotional processing will provide a better and more accurate background, because as it stands, this information is not highlighted in the text. In this regard, I would suggest to add more information about neural substrates underlying fear processing in humans (https://doi.org/10.3389/fnbeh.2022.998714), as well as functional abnormalities of specific brain regions (i.e., prefrontal cortex), and on related effects on patients’ cognitive impairments (https://doi.org/10.1111/psyp.14122).
• fMRI paradigm: I was wondering whether the number of experimental trials is enough to achieve an acceptable level of statistical power.
• Results: I suggest rewriting this section more accurately. To properly present experimental findings, I think that authors should provide full statistical details (like degree of freedom or post-hoc utilized), to ensure in-depth understanding and replicability of the findings. Also, in my opinion, it is necessary for the authors to present their findings using summary tables.
• Although not mandatory, I believe that a proper and defined ‘Conclusions’ paragraph would be useful to properly address thoughtful as well as in-depth considerations by the authors. Authors should make an effort, trying to explain the theoretical implication as well as the translational application of their research.
• In according to the previous comment, I would ask the authors to include a proper and defined ‘Limitations and future directions’ section before the end of the manuscript, in which authors can describe in detail and report all the technical issues brought to the surface.
• Tables and Figures: According to the Journal’s guidelines, please provide a short explanatory caption for the table within the text. Also, please add error bars in all the graphs.
• References: Authors should consider revising the bibliography, as there are several incorrect citations. Indeed, according to the Journal’s guidelines, they should provide the abbreviated journal name in italics, the year of publication in bold, the volume number in italics for all the references.
I hope that, after these careful revisions, this paper can meet the Journal’s high standards for publication.
I am available for a new round of revision of this paper. I declare no conflict of interest regarding this manuscript.
Best regards,
Reviewer
Author Response
General remarks: The present research article by Günther, entitled ‘Cognitive avoidance is associated with decreased brain responsiveness to threat distractors under high perceptual load’ is a well-written and useful summary on the current status of knowledge on neural underpinnings of copying strategies to inhibit threat processing.
The main strength of this manuscript is that it addresses an interesting and timely question, describing how the tendency to divert one’s attention away from distressing stimuli may be a valuable predictor of diminished automatic neural responses to threat in several cortical and subcortical areas.
In general, I think the idea of this article is really interesting and the authors’ fascinating observations on this timely topic may be of interest to the readers of Brain Sciences. However, some comments, as well as some crucial evidence that should be included to support the author’s argumentation, needed to be addressed to improve the quality of the manuscript, its adequacy, and its readability prior to the publication in the present form, in particular reshaping parts of the Introduction and Methods sections by adding more evidence and theoretical constructs.
Comment 1: Abstract: According to the Journal’s guidelines, the abstract should be a total of about 200 words maximum. Please correct the actual one.
Our response: The abstract of our revised manuscript comprises one paragraph consisting of 200 words. According to the author guidelines of Brain Sciences the abstract should be a total of 200 words maximum.
Comment 2: A graphical abstract that will visually summarize the main findings of the manuscript is highly recommended.
Our response: This is a good point. We created a graphical abstract where our main results are summarized.
Comment 3: In general, I recommend authors to use more references to back their claims, especially in the Introduction of this article, which I believe is lacking. Thus, I recommend the authors to attempt to expand the topic of their article, as the bibliography is too concise. Nevertheless, I believe that less than 50 articles are too low for a research article. Therefore, I suggest the authors to focus their efforts on researching relevant literature: in my opinion, adding more citations will help to provide better and more accurate background to this study.
Our response: We agree with the reviewer that in a typical research article more than 50 references should be cited. Our research paper comprises one hundred and ten references (articles, book chapters, and books). In our view, the literature cited in our paper is highly relevant and provides a good theoretical background for introducing our research question.
Comment 4: Introduction: The ‘Introduction’ section is well-written and nicely presented, with a good balance of descriptive text and information about neural processing of threating emotions. Nevertheless, I believe that more information about neural bases of emotional processing will provide a better and more accurate background, because as it stands, this information is not highlighted in the text. In this regard, I would suggest to add more information about neural substrates underlying fear processing in humans (https://doi.org/10.3389/fnbeh.2022.998714), as well as functional abnormalities of specific brain regions (i.e., prefrontal cortex), and on related effects on patients’ cognitive impairments (https://doi.org/10.1111/psyp.14122).
Our response: Indeed, the inclusion of literature on neural threat processing in our introduction may be of interest for the readers. In our revised text, we cited the mentioned reference on the impact of emotions on action control and refer the reader to a review and meta-analyses where neural bases of emotion (threat) processing are summarized [p. 3, 4th paragraph]. In our opinion, our introduction is relatively extensive with 1,500 words. Therefore, we would refrain from introducing literature on abnormal emotion processing in clinical samples, since our study investigated personality constructs in healthy subjects. We hope the reviewer agrees to our decision.
Comment 5: fMRI paradigm: I was wondering whether the number of experimental trials is enough to achieve an acceptable level of statistical power.
Our response: In our experience, a trial number of n >= 20 per condition is adequate and quite common in fMRI research (e.g., 10.1016/j.neuroscience.2019.03.059; 10.1093/scan/nsu104), in particular in studies using a perceptual load task (e.g., 10.1016/j.pnpbp.2017.04.007; 10.1016/j.biopsycho.2014.06.004; 10.1093/cercor/bhl070). In the present experiment, we have used 24 trials per condition. In our revised limitation section [p. 15, 2nd paragraph] we have included the notion that an increased trial number may enhance the statistical power, but might result in habituation effects.
Comment 6: Results: I suggest rewriting this section more accurately. To properly present experimental findings, I think that authors should provide full statistical details (like degree of freedom or post-hoc utilized), to ensure in-depth understanding and replicability of the findings. Also, in my opinion, it is necessary for the authors to present their findings using summary tables.
Our response: The reviewer is right to recommend a report of degrees of freedom. In our revised results section, we added the respective information to our correlation results [bottom of p.6] and in the caption of Table 1 [p.4]. Regarding our neuroimaging results, we summarize the usual statistics (hemisphere, peak t-value, p-values at peak level and cluster level, cluster size, and MNI coordinates) for the relevant brain areas in our detailed Table 2 and 3 [p. 7ff].
Comment 7: Although not mandatory, I believe that a proper and defined ‘Conclusions’ paragraph would be useful to properly address thoughtful as well as in-depth considerations by the authors. Authors should make an effort, trying to explain the theoretical implication as well as the translational application of their research.
Our response: The reviewer is right to recommend a conclusion section in our text. We hope the reviewer agrees to our decision to keep this part concise and to avoid excessive speculations. In the revised conclusion part [p. 15] our main findings are summarized and we give a brief prospect of their potential clinical relevance. In a previous more comprehensive paragraph [p. 14, 3rd paragraph], we provide theoretical implications of our findings regarding a vulnerability to non-clinical and clinical anxiety. Since most of these considerations are highly speculative, we would refrain from extending this section.
Comment 8: In according to the previous comment, I would ask the authors to include a proper and defined ‘Limitations and future directions’ section before the end of the manuscript, in which authors can describe in detail and report all the technical issues brought to the surface.
Our response: The reviewer is right to request a limitation and future direction section. In our revised discussion [p. 15, 2nd paragraph] we included a separate section were we provide several limitations and suggestions for future research.
Comment 9: Tables and Figures: According to the Journal’s guidelines, please provide a short explanatory caption for the table within the text. Also, please add error bars in all the graphs.
Our response: Following the author guidelines we inserted all figures and tables into the main text close to their first citation. Our figures and tables have all short explanatory titles and captions. In our revised manuscript, no graphs are presented.
Comment 10: References: Authors should consider revising the bibliography, as there are several incorrect citations. Indeed, according to the Journal’s guidelines, they should provide the abbreviated journal name in italics, the year of publication in bold, the volume number in italics for all the references.
Our response: In our revised manuscript, we have modified the citation style and bibliography format according to the author guidelines.
Round 2
Reviewer 2 Report
The authors did an excellent job clarifying all the questions I have raised in my previous round of review. Currently, this paper is a well-written, timely piece of research that described how the tendency to divert one’s attention away from distressing stimuli may be a valuable predictor of diminished automatic neural responses to threat in several cortical and subcortical areas.
Overall, this is a timely and needed work. It is well-researched and nicely written, therefore I believe that this paper does not need a further revision.
I am always available for other reviews of such interesting and important articles. Thank You for your work,
Reviewer